# On the Discrimination Risk of Mean Aggregation Feature Imputation in Graphs

**Arjun Subramonian**
UCLA
arjunsub@cs.ucla.edu

**Kai-Wei Chang**
UCLA
kwchang@cs.ucla.edu

**Yizhou Sun**
UCLA
yzsun@cs.ucla.edu

## Abstract

In human networks, nodes belonging to a marginalized group often have a dispro-portionate rate of unknown or missing features. This, in conjunction with graph structure and known feature biases, can cause graph feature imputation algorithms to predict values for unknown features that make the marginalized group's feature values more distinct from the the dominant group's feature values than they are in reality. We call this distinction the **discrimination risk**. We prove that a higher discrimination risk can amplify the unfairness of a machine learning model ap-plied to the imputed data. We then formalize a general graph feature imputation framework called mean aggregation imputation and theoretically and empirically characterize graphs in which applying this framework can yield feature values with a high discrimination risk. We propose a simple algorithm to ensure mean aggregation-imputed features provably have a low discrimination risk, while min-imally sacrificing reconstruction error (with respect to the imputation objective). We evaluate the fairness and accuracy of our solution on synthetic and real-world credit networks.

## 1 Introduction

Machine learning (ML) methods [1, 2, 3, 4, 5] have been successfully applied to graph-structured data such as social networks, product graphs, and molecules [6, 7] to aid in important problems like content recommendation, product recommendation, and molecular property prediction [8, 9, 7, 10, 11]. However, many methods rely on fully-observed features for each node, which are not available for privacy reasons, as a consequence of exclusionary data collection practices, or due to the high expenses involved in feature annotation [12, 13, 14]. As a result, the development of algorithms to leverage a graph's structure and known node feature values to impute unknown or missing features has emerged as an important research area [15, 16].

In human networks, nodes belonging to a marginalized group (e.g., on the basis of race, gender, disability, etc.) often have a disproportionate rate of unknown features compared to the dominant group because marginalized communities may be more reluctant to share their data, annotators erase their data, and they are sidelined in data collection [17, 18, 19, 20]. Furthermore, node neighborhoods are often associated with group membership [21, 22], especially in homophilic graphs where nodes belonging to the same group have a higher likelihood of being connected [23]. Homophily can be due to social stratification [24] or the limited collection of inter-links (i.e., edges between nodes belonging to different groups) [23, 7]. Moreover, known feature values can be tainted and proxies for group membership [18, 25]. Hence, even if graph feature imputation algorithms do not have direct access to the group membership of a node, these algorithms are influenced by unknown feature rate disparities, graph structure, and known feature values. In fact, they can predict values for unknown features that cause the the marginalized group's feature values to be more distinct from the dominant group's feature values than they are in reality.

36th Conference on Neural Information Processing Systems (NeurIPS 2022).

To illustrate this phenomenon, let's consider an automated candidate screening system based on the job applicant network shown in Figure 1, where nodes represent applicants and edges between applicants indicate that they have similar past work experiences. Each node has one feature: the number of years that the applicant has previously worked. Furthermore, each node belongs to one of two groups, the disabled community $Q$ or the able-bodied community $R$[1]. All the nodes in $R$ have a known feature value of 5 years, while all the nodes in $Q$ have unknown feature values; however, in reality, all the nodes in $Q$ also have a feature value of 5 years. Additionally, all the nodes in $Q$ are connected to each other but have few links (if any) to nodes in $R$ because of systemic barriers including hiring discrimination and a lack of accommodations [27]. Consequently, after applying the graph feature imputation algorithm Feature Propagation [16], because of the disparate rates of unknown features between the groups and structure of the graph, the nodes in $Q$ will have, on average, feature values that are more distinct from the feature values of the nodes in $R$ than the ground truth.

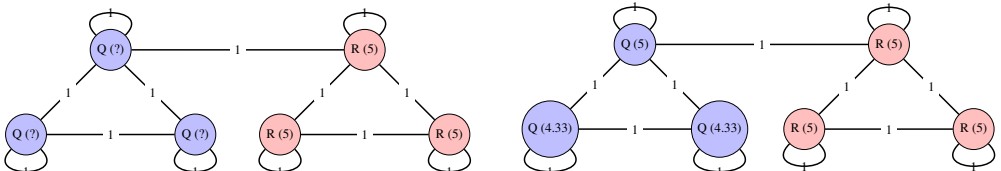

Figure 1: A job applicant network. After applying Feature Propagation [16], the nodes in $Q$ will have feature values that are more distinct from the feature values of the nodes in $R$ than the ground truth.

We call this distinction in imputed feature values between the marginalized and dominant groups the **discrimination risk**. In this paper, we present a theoretically-justified formulation of the discrimination risk of imputed features. We further prove that a higher discrimination risk can amplify the unfairness of a ML model applied to the imputed data, which can be especially dangerous, unethical, and illegal in high-stakes applications like automated candidate screening and loan approval [28, 29, 30, 31]. For instance, in the automated candidate screening example, a model applied to the imputed data may more easily learn to identify and reject disabled job applicants because they appear to have fewer years of work experience than able-bodied applicants, thereby reinforcing systemic discrimination against the disabled community [27].

We also formalize a general graph feature imputation framework called **mean aggregation imputation** that encompasses common diffusion-based imputation algorithms in the literature. Subsequently, we theoretically and empirically characterize graphs in which applying mean aggregation feature imputation can yield a high discrimination risk, without making any assumptions about the underlying distributions of unknown features or graph structure. To the best of our knowledge, we are the first to study the effect of graph feature imputation on the fairness of models. This is challenging because we must consider biases stemming from graph structure.

Furthermore, we propose a simple algorithm to ensure mean aggregation-imputed features provably have a low discrimination risk, while minimally sacrificing reconstruction error (with respect to the imputation objective). We do so by viewing mean aggregation imputation through the lens of gradient descent and projecting imputed feature values onto the feasible space of feature values with low discrimination risk. We empirically evaluate the fairness and accuracy of our solution on synthetic and real-world credit networks, finding that it improves fairness without a significant loss in reconstruction error on the synthetic datasets but doesn't improve fairness on the real-world datasets. We close by discussing the limitations of our solution.

## 2 Related work

**Feature imputation** Feature imputation algorithms leverage known feature values to predict unknown feature values (and sometimes update known feature values). For example, unknown feature values may be filled as the mean of known values [16]. However, more intricate feature imputation methods have been proposed in the ML, statistics, and epidemiology literature, with popular approaches including matrix completion [32, 33, 34, 35], nearest neighbors [36, 37], multiple imputation via conditional models [38, 39, 40], and causal inference [41, 42]. Notably, while feature imputation may be applied to data with unknown feature values prior to the data being passed to a ML model,

---

[1]In reality, disability is a fluid and complex identity that should not be reduced to a binary [26].

feature imputation is distinct from label prediction with missing data, wherein models work directly with unknown feature values [43, 44, 45, 46, 47, 48, 49, 50]; we do not consider the latter paradigm in this paper. Works have extended feature imputation approaches for tabular data to incorporate graph structure [51, 52]. Graph feature imputation using deep learning methods is also gaining traction [53, 54, 15]. However, [16] proposes a non-neural diffusion-based approach called Feature Propagation that imputes graph features by minimizing the graph's Dirichlet energy.

**Fairness of missing data and feature imputation** Works have theoretically and empirically investigated the impact of missing data [55, 56, 19, 57, 58] and feature imputation on the fairness of ML models [56, 59, 60, 61, 30, 62]. These works consistently find that missing data can amplify biases, and some show that in practice, feature imputation can yield less unfair (relative to excluding missing data) but nevertheless discriminatory model outcomes. However, these works only study tabular data and do not consider biases that emerge from graph structure [20, 22, 63]. Furthermore, they often adapt models to directly work with missing data rather than mitigate the unfairness of feature imputation itself. Despite the prevalence of graph feature imputation methods, to the best of our knowledge, there is no research on their influence on the fairness of models. Some works have explored fairness constraints in semi-supervised settings [64, 65], but they assume that node features are entirely available, which is not the case in feature imputation.

**Fair graph machine learning** We focus on group fairness, which ensures model predictions exhibit some form of parity between different groups [66, 21, 67, 29, 68, 69]. Works have studied (amongst other fairness formulations) statistical parity, wherein a model predicts the positive outcome at the same rate for different groups, and equalized odds, in which the accuracy of model predictions is equivalent across different groups [70]. In the automated candidate screening example, statistical parity would imply that all candidates have an equivalent opportunity to pass screening regardless of group membership, while equalized odds would mean that, regardless of group membership (i.e., $\mathbb{P}(Z|S = Q) = \mathbb{P}(Z|S = R)$), candidates are classified with an equivalent accuracy by the automated screening model (i.e., $\mathbb{P}(Z|Y, S = Q) = \mathbb{P}(Z|Y, S = R)$). When ground-truth labels are tainted (e.g., the screening system is trained with sexist hiring data), we may prefer statistical parity to equal opportunity. While our theoretical study of discrimination risk is aligned with statistical parity, we empirically explore the effect of mean aggregation imputation with a lower discrimination risk on both the demographic parity and equalized odds of models. Mechanisms for improving group fairness have been categorized into pre-processing [22], training-time [21, 23, 71], and post-processing [72, 73]. Our work is similar to pre-processing, as we seek to lower the discrimination risk of imputed training data towards improving model fairness. There exist many works on modifying graph structure to mitigate topology-induced biases [73, 74, 22, 71, 75, 76]. However, because graph semantics (especially for large graphs) are difficult to interpret, it is unclear if the solutions that these papers propose preserve the semantics of the original graph. On the other hand, our work does not modify graph structure and instead optimally transforms imputed features to have a low discrimination risk. [77] investigates the fairness of graph neural networks in the presence of limited group membership information but does not consider imputation; in contrast, our work assumes group membership is fully available but features are not.

## 3 Discrimination risk and model unfairness

To understand how feature imputation could amplify the unfairness of a ML model, we now present a theoretically-justified metric called the discrimination risk, which applies beyond the setting of graphs and is agnostic to model architecture and labels. (We explore discrimination risk in the context of graphs in Section 5.) Suppose we have an arbitrary data distribution $\mathcal{D}$ and two groups: a marginalized group $Q$ and a dominant group $R$. For any data instance $(x, y, s) \sim \mathcal{D}$, let $x \in \mathcal{X}$ be the $d$-dimensional feature values of the instance (where $x_i$ denotes the $i$-th entry of $x$), let $y \in \mathcal{Y}$ be its label, and let $s \in \{Q, R\}$ be its group membership. We assume that we can observe the group membership of any data instance and that no instance can belong to both the marginalized and dominant groups.

**Definition 1 (Discrimination Risk)** We define the discrimination risk of $\mathcal{D}$ as:

$$\mathcal{R}_{\mathcal{D}} = \left\| \mathbb{E}_{(x,y,s)\sim\mathcal{D}}[x|s = Q] - \mathbb{E}_{(x,y,s)\sim\mathcal{D}}[x|s = R] \right\|_{\infty}, \tag{1}$$

We now explore the relevance of discrimination risk to model unfairness. Let $\mathcal{D}'$ be the data distribution with ground-truth features and $\mathcal{D}$ be the distribution with imputed features. We will show that if $\mathcal{R}_{\mathcal{D}} > \mathcal{R}_{\mathcal{D}'}$, imputation may amplify model unfairness.

To be concrete, let's again consider the example of the automated candidate screening system. Let $\mathbb{P}(S \in \{Q, R\})$ be the distribution over the group membership of job applicants. Furthermore, let $\mathbb{P}(X' \in \mathcal{X})$ be the distribution over the ground-truth feature values of applicants (e.g., number of years previously worked, highest degree, etc.) In contrast, let $\mathbb{P}(X \in \mathcal{X})$ be the distribution over imputed feature values of applicants. Let $\mathbb{P}(Y \in \mathcal{Y})$ be the distribution over ground-truth applicant labels (e.g., whether an applicant should be hired, a screening score for an applicant, etc.) Now, let $h'$ be a model trained on samples from $\mathcal{D}'$ (without direct access to $S$), and $h$ be another model trained on samples from $\mathcal{D}$ (without direct access to $S$). Additionally, let $\mathbb{P}(Z' \in \mathcal{Y})$ be the distribution over the predictions of $h'$ on instances sampled from $\mathbb{P}(X')$, and let $\mathbb{P}(Z \in \mathcal{Y})$ be the distribution over the predictions of $h$ on instances sampled from $\mathbb{P}(X)$.

We have the dependencies $S \rightarrow X'$ and $S \rightarrow X$ because disability affects the number of years previously worked by a job applicant. Additionally, $S \rightarrow X$ because $S$ can influence the feature imputation algorithm (as illustrated in Figure 1). Furthermore, we assume that $h'$ and $h$ have access to the feature values of an applicant, but not the applicant's group membership, and that the association of $S$ with $Z'$ through $Y$ can be fully explained by $X'$. Thus, $Z'$ is conditionally independent of $S$ given $X'$. Similarly, we assume that the association of $S$ with $Z$ through $Y$ can be fully explained by $X$, so $Z$ is conditionally independent of $S$ given $X$.

Because we are interested in how feature imputation may amplify model unfairness, we investigate when the statistical parity or total variation distance $d_{TV}(\mathbb{P}(Z|S = Q), \mathbb{P}(Z|S = R)) > d_{TV}(\mathbb{P}(Z'|S = Q), \mathbb{P}(Z'|S = R))$ [78], where $\mathbb{P}(Z|S = Q)$ is the prediction distribution conditioned on group-$Q$ membership. $d_{TV}(A, B)$ measures the distance between two probability distributions $A$ and $B$ as $\sup_{x \in \mathcal{F}} |A(x) - B(x)|$. Intuitively, $d_{TV}(\mathbb{P}(Z|S = Q), \mathbb{P}(Z|S = R))$ captures how much the prediction distribution of $h$ absolutely differs between $Q$ and $R$, and thus it quantifies the (statistical parity) unfairness of $h$. We now discuss the relationships of $d_{TV}(\mathbb{P}(Z'|S = Q), \mathbb{P}(Z'|S = R))$ to $d_{TV}(\mathbb{P}(X'|S = Q), \mathbb{P}(X'|S = R))$ and $d_{TV}(\mathbb{P}(Z|S = Q), \mathbb{P}(Z|S = R))$ to $d_{TV}(\mathbb{P}(X|S = Q), \mathbb{P}(X|S = R))$. We begin with the following lemma from [79].

**Lemma 1 (Corollary 17 from [79])** By the Data Processing Inequality, $d_{TV}(\mathbb{P}(Z'|S = Q), \mathbb{P}(Z'|S = R)) \leq d_{TV}(\mathbb{P}(X'|S = Q), \mathbb{P}(X'|S = R))$ and $d_{TV}(\mathbb{P}(Z|S = Q), \mathbb{P}(Z|S = R)) \leq d_{TV}(\mathbb{P}(X|S = Q), \mathbb{P}(X|S = R))$.

Please refer to Section A.1 for the proof of Lemma 1. At a high level, Lemma 1 states that the statistical parity unfairness of a model is upper-bounded by the statistical parity distance of the feature distributions between the groups. We now use Lemma 1 to prove the following theorem that connects $d_{TV}(\mathbb{P}(Z'|S = Q), \mathbb{P}(Z'|S = R))$ to $\mathcal{R}_{\mathcal{D}'}$ and $d_{TV}(\mathbb{P}(Z|S = Q), \mathbb{P}(Z|S = R))$ to $\mathcal{R}_{\mathcal{D}}$.

**Theorem 1** Suppose $\mathbb{P}(X'|S = Q) = \mathcal{N}(\mu'_Q, \Sigma'_Q); \mathbb{P}(X'|S = R) = \mathcal{N}(\mu'_R, \Sigma'_R); \mathbb{P}(X|S = Q) = \mathcal{N}(\mu_Q, \Sigma_Q);$ and $\mathbb{P}(X|S = R) = \mathcal{N}(\mu_R, \Sigma_R)$. We then get the following bounds:

$$d_{TV}(\mathbb{P}(Z'|S = Q), \mathbb{P}(Z'|S = R)) \in \left[ \frac{1}{\frac{4 \cdot \max\{\lambda_{max}(\Sigma'_Q), \lambda_{max}(\Sigma'_R)\}}{C' \cdot \mathcal{R}^2_{\mathcal{D}'}} + 1}, \sqrt{1 - \sqrt{\frac{\det \Sigma'_Q}{\det \Sigma'_R} \cdot e^{-\frac{C' \cdot \mathcal{R}^2_{\mathcal{D}'}}{\lambda_{min}(\Sigma'_R)} - tr(\Sigma'^{-1}_R \Sigma'_Q) + d}}} \right];$$

$$d_{TV}(\mathbb{P}(Z|S = Q), \mathbb{P}(Z|S = R)) \in \left[ \frac{1}{\frac{4 \cdot \max\{\lambda_{max}(\Sigma_Q), \lambda_{max}(\Sigma_R)\}}{C \cdot \mathcal{R}^2_{\mathcal{D}}} + 1}, \sqrt{1 - \sqrt{\frac{\det \Sigma_Q}{\det \Sigma_R} \cdot e^{-\frac{C \cdot \mathcal{R}^2_{\mathcal{D}}}{\lambda_{min}(\Sigma_R)} - tr(\Sigma^{-1}_R \Sigma_Q) + d}}} \right],$$

where $0 \leq C' \leq d$ and $0 \leq C \leq d$. Please refer to Section A.2 for the proof of Theorem 1.

This result suggests that minimizing $\mathcal{R}_{\mathcal{D}}$ can minimize the unfairness of the model $h$ applied to the imputed data. Furthermore, it is possible that $\mathcal{R}_{\mathcal{D}} > \mathcal{R}_{\mathcal{D}'}$, in which case feature imputation may amplify the unfairness of a model. While we leverage strong generative assumptions in Theorem 1, it is plausible that $X'|S = Q$ and $X'|S = R$ are normally-distributed. Furthermore, mean aggregation imputation (Section 4) produces approximately normal $X|S = Q$ and $X|S = R$ (by the Central Limit Theorem). We additionally note that the lower bounds do not require the feature values to be normally distributed; the bounds only assume their distributions have finite covariance. Finally, for arbitrary distributions, matching even an infinite number of moments is not sufficient to bound their distance [80].

While $\mathcal{R}_{\mathcal{D}}$ applies beyond graphs and is agnostic to model architecture and labels, in practice, it is important to consider model complexity and task context. We also add that $\mathcal{R}_{\mathcal{D}}$ risk bears resemblance to the Average Treatment Effect studied in causality [81]. Moreover, [59] proposes a metric also

called discrimination risk which quantifies how much the deviation of imputed feature values from the ground-truth feature values differs across groups; this quantity is aligned with the equalized odds of a model applied to the imputed data. In contrast, $\mathcal{R}_\mathcal{D}$ simply measures how much imputed features differ across groups, enabling its computation when ground-truth feature values are not available, and is aligned with the statistical parity of a model. We leave extending our definition of discrimination risk to other formulations of fairness as future work [70].

## 4 Graph feature imputation

We would like to investigate the discrimination risk of graph feature imputation. Prior to doing so, we present a general framework called mean aggregation imputation that encompasses common non-neural diffusion-based graph feature imputation algorithms in the literature.

Suppose we have an undirected weighted homogeneous graph $G = (V, E)$. Each node has $d$ features, hence the node feature matrix $X \in \mathbb{R}^{N \times d}$, where $N = |V|$ (i.e., the cardinality of $V$). For simplicity, we assume that $d = 1$. The feature value is unknown for some nodes in $G$ (denoted as the set $U$), and known for others (denoted as the set $K$). The feature value of each node is either known or unknown (i.e., $U \cup K = V$ and $U \cap K = \emptyset$). Let $X_S$ refer to the feature values of the nodes in set $S$. Assume without loss of generality that $X = \begin{bmatrix} X_K \\ X_U \end{bmatrix}$. Furthermore, let $A \in \mathbb{R}^{N \times N}$ denote the weighted adjacency matrix of $G$, where $A_{ij}$ is the nonnegative weight corresponding to the edge from node $j$ to node $i$. Additionally, let $A_{S_1 S_2}$ denote the submatrix of $A$ with rows belonging to the nodes in set $S_1$ and columns belonging to the nodes in set $S_2$. Let $A := \begin{bmatrix} A_{KK} & A_{KU} \\ A_{UK} & A_{UU} \end{bmatrix}$. $D$ is the diagonal degree matrix, i.e., $D_{ii} = \sum_{j=1}^{N} A_{ij}$ and $D := \begin{bmatrix} D_K & 0 \\ 0 & D_U \end{bmatrix}$.

**Definition 2 (Mean Aggregation Feature Imputation)** Denote the feature values at iteration $t$ of mean aggregation feature imputation as $X^{(t)}$. Furthermore, let $X_S^{(t)}$ refer to the feature values of the nodes in set $S$ at iteration $t$. Then, at each iteration $t$:

$$MX^{(t+1)} := \phi(MX^{(t)}) = \begin{bmatrix} \beta I_{|K|} & 0 \\ 0 & I_{|U|} \end{bmatrix} TMX^{(t)} + \begin{bmatrix} (1-\beta)I_{|K|} & 0 \\ 0 & 0 \end{bmatrix} MX^{(0)}, \quad (2)$$

where $M : \mathbb{R}^N \to \mathbb{R}^N$ is a diagonal invertible map, $T \in \mathbb{R}^{N \times N}$ is a right-stochastic matrix, and $\beta \in [0, 1]$ is a regularization hyperparameter.

When $d > 1$, we apply mean aggregation feature imputation to each channel independently[2], and it only works with continuous (not discrete) features. Mean aggregation encompasses common graph feature imputation methods, including **Global Mean** (predicts unknown feature values as the uniform mean of known feature values), **Neighbor Mean** (predicts unknown feature values as the degree-weighted mean of the known feature values for neighboring nodes), **Feature Propagation** (predicts unknown feature values that minimize the Dirichlet energy of the graph while preserving known feature values), and **Graph Regularization** (predicts feature values via a smoothness constraint and a fitting constraint for the known features). For proofs, refer to Section A.3. Despite Neighbor Mean, Feature Propagation, and Graph Regularization being intended for homophilic graphs [16], mean aggregation feature imputation encompasses algorithms that could perform well on heterophilic graphs as well with an appropriate choice of $T$ [16]. We also note that $T$ cannot be $A$, as this might cause feature values to explode over multiple iterations.

## 5 Discrimination risk of mean aggregation feature imputation

To understand how mean aggregation feature imputation may amplify the unfairness of a ML model, we theoretically characterize graphs in which mean aggregation imputation increases the discrimination risk, without making assumptions about the underlying distributions of unknown features or graph structure.

We begin by defining new notation. In particular, we first focus on the case of a single feature (i.e., $d = 1$), and extend our analysis to the case $d > 1$ in Section A.5. Let $X_v$ be the feature of a node $v$.

---

[2]Incorporating associations between features is a promising direction of research.

Let $Q$ denote the set of nodes that belong to the marginalized group, and $R$ denote the set of nodes that belong to the dominant group. We assume that $Q \cup R = V$ and $Q \cap R = \emptyset$. We define the discrimination risk after $t$ iterations of mean aggregation imputation as:

$$\mathcal{R}^{(t)} := \left| \mathbb{E}_{q \sim Q}[X_q^{(t)}] - \mathbb{E}_{r \sim R}[X_r^{(t)}] \right|, \tag{3}$$

where the expectations are taken uniformly over the nodes in each set. Now, define $\tilde{X} := MX$. Then, a modified version of the discrimination risk after $t$ iterations of mean aggregation imputation is $\tilde{\mathcal{R}}^{(t)} := \left| \mathbb{E}_{q \sim Q}[\tilde{X}_q^{(t)}] - \mathbb{E}_{r \sim R}[\tilde{X}_r^{(t)}] \right|$. In Theorem 2, we bound the discrimination risk $\tilde{\mathcal{R}}^{(t)}$ of the imputed features with respect to $\tilde{\mathcal{R}}^{(0)}$. We bound $\tilde{\mathcal{R}}^{(t)}$ rather than $\mathcal{R}^{(t)}$ for simplicity; however, we empirically validate that the bound properties also hold for $\mathcal{R}^{(t)}$ in Section 7. Define $\tilde{\mu}_Q^{(t)} := \mathbb{E}_{q \sim Q}[\tilde{X}_q^{(t)}]$ and $\tilde{\mu}_R^{(t)} := \mathbb{E}_{r \sim R}[\tilde{X}_r^{(t)}]$. Furthermore, let $\tilde{\sigma}^{(t)}$ denote the maximal deviation of the feature values at iteration $t$, i.e., $\forall q_1 \in Q, |\tilde{X}_{q_1}^{(t)} - \tilde{\mu}_Q^{(t)}| \leq \max_{q_2 \in Q} |\tilde{X}_{q_1}^{(t)} - \tilde{X}_{q_2}^{(t)}| \leq \tilde{\sigma}^{(t)}$ and $\forall r_1 \in R, |\tilde{X}_{r_1}^{(t)} - \tilde{\mu}_R^{(t)}| \leq \max_{r_2 \in R} |\tilde{X}_{r_1}^{(t)} - \tilde{X}_{r_2}^{(t)}| \leq \tilde{\sigma}^{(t)}$. Additionally, define $T_{S_1 \to S_2} := \sum_{b \in S_2} \sum_{a \in S_1} T_{ba}$.

**Theorem 2** Let the contraction coefficient $\alpha := \left| 1 - \frac{T_{R \to Q \cap U} + \beta T_{R \to Q \cap K}}{|Q|} - \frac{T_{Q \to R \cap U} + \beta T_{Q \to R \cap K}}{|R|} \right|$. Then, $\alpha \leq 1$, and:

$$\max \left\{ \alpha^t \tilde{\mathcal{R}}^{(0)} - 2 \left( \sum_{j=0}^{t-1} \alpha^j \right) \tilde{\sigma}^{(0)}, 0 \right\} \leq \tilde{\mathcal{R}}^{(t)} \leq \alpha^t \tilde{\mathcal{R}}^{(0)} + 2 \left( \sum_{j=0}^{t-1} \alpha^j \right) \tilde{\sigma}^{(0)}$$

$$\alpha < 1 \implies \lim_{t \to \infty} \tilde{\mathcal{R}}^{(t)} \leq \frac{2 \tilde{\sigma}^{(0)}}{1 - \alpha}.$$

Please refer to Section A.4 for a proof of Theorem 2. Theorem 2 shows that the bounds on the discrimination risk contract more slowly (with more iterations of mean aggregation feature imputation) with a larger $\alpha$. Furthermore, the upper bound on the discrimination risk is larger with a larger $\alpha$ and may depend on the initial unknown feature values.

### 5.1 Analysis of Theorem 2

Theorem 2 allows for interesting interpretations of how graph properties like the rate of unknown features, group size, and graph structure affect the discrimination risk of mean aggregation imputation. Below, we successively vary each property (holding the other properties constant) and investigate its impact on $\alpha$, and in turn the discrimination risk. We focus on Feature Propagation [16], but our analysis may be easily extended to other mean aggregation imputation algorithms. We assume for simplicity that all edges have a weight of 1.

**Unknown feature rates** *A low unknown feature rate for both groups or disparate unknown feature rates across the groups can increase $\alpha$, and thus the discrimination risk of mean aggregation-imputed features.* Suppose the intra-link rate $\mathbb{P}((u,v) \in E | u \in Q, v \in Q) = \mathbb{P}((u,v) \in E | u \in R, v \in R) = \frac{1}{2}$ and inter-link rate $\mathbb{P}((u,v) \in E | u \in Q, v \in R) = \mathbb{P}((u,v) \in E | u \in R, v \in Q) = \frac{1}{2}$. Furthermore, assume equal (relative) group sizes, i.e., $\frac{|Q|}{N} = \frac{|R|}{N} = \frac{1}{2}$. Then, $\frac{T_{R \to Q \cap U} + \beta T_{R \to Q \cap K}}{|Q|} = \frac{T_{R \to Q \cap U}}{N/2} = \frac{\sum_{q \in Q \cap U} \sum_{r \in R} T_{qr}}{N/2} = \frac{\sum_{q \in Q \cap U} \sum_{r \in R} D_{qq}^{-1} A_{qr}}{N/2}$. By decomposition, $D_{qq} = \sum_{u \in Q} A_{qu} + \sum_{v \in R} A_{qv} = \frac{1}{2}|Q| + \frac{1}{2}|R| = \frac{N}{2}$. Therefore, $\frac{T_{R \to Q \cap U} + \beta T_{R \to Q \cap K}}{|Q|} = \frac{\sum_{q \in Q \cap U} \sum_{r \in R} A_{qr}}{N^2/4} = \frac{\frac{1}{2}(|Q \cap U| \times |R|)}{N^2/4} = \frac{\frac{1}{2}(\mathbb{P}(v \in U | v \in Q) \cdot |Q| \times |R|)}{N^2/4} = \frac{1}{2}\mathbb{P}(v \in U | v \in Q)$, where $\mathbb{P}(v \in U | v \in Q)$ is the unknown feature rate for group $Q$. Similarly, $\frac{T_{Q \to R \cap U} + \beta T_{Q \to R \cap K}}{|R|} = \frac{1}{2}\mathbb{P}(v \in U | v \in R)$. Thus, $\alpha = |1 - \frac{1}{2}\mathbb{P}(v \in U | v \in Q) - \frac{1}{2}\mathbb{P}(v \in U | v \in R)|$. This aligns with [60]'s finding that imputation fairness can be influenced by the imbalance of feature missingness across groups, although [60] studies equalized odds rather than statistical parity fairness.

**Group sizes** *Group size alone may not affect $\alpha$ or the discrimination risk of mean aggregation-imputed features.* Suppose the intra-link rate $\mathbb{P}((u,v) \in E | u \in Q, v \in Q) = \mathbb{P}((u,v) \in E | u \in R, v \in R) = \frac{1}{2}$ and inter-link rate $\mathbb{P}((u,v) \in E | u \in Q, v \in R) = \mathbb{P}((u,v) \in E | u \in R, v \in Q) = \frac{1}{2}$.

Furthermore, assume equal unknown feature rates, i.e., $\mathbb{P}(v \in U | v \in Q) = \mathbb{P}(v \in U | v \in R) = \frac{1}{2}$. Then, $\frac{T_{R \to Q \cap U} + \beta T_{R \to Q \cap K}}{|Q|} = \frac{\sum_{q \in Q \cap U} \sum_{r \in R} T_{qr}}{|Q|} = \frac{\sum_{q \in Q \cap U} \sum_{r \in R} D_{qq}^{-1} A_{qr}}{|Q|}$. $D_{qq} = \frac{N}{2}$. Therefore, $\frac{T_{R \to Q \cap U} + \beta T_{R \to Q \cap K}}{|Q|} = \frac{\sum_{q \in Q \cap U} \sum_{r \in R} A_{qr}}{|Q| \cdot N/2} = \frac{\frac{1}{2}(|Q \cap U| \times |R|)}{|Q| \cdot N/2} = \frac{\frac{1}{2}(\frac{1}{2}|Q| \times |R|)}{|Q| \cdot N/2} = \frac{1}{2} \cdot \frac{|R|}{N}$. Similarly, $\frac{T_{Q \to R \cap U} + \beta T_{Q \to R \cap K}}{|R|} = \frac{1}{2} \cdot \frac{|Q|}{N}$. Thus, $\alpha = |1 - \frac{1}{2} \cdot \frac{|R|}{N} - \frac{1}{2} \cdot \frac{|Q|}{N}| = \frac{1}{2}$.

**Graph structure** *A low inter-link to intra-link ratio can increase $\alpha$ and the discrimination risk of mean aggregation-imputed features.* Suppose we have equal unknown feature rates, i.e., $\mathbb{P}(v \in U | v \in Q) = \mathbb{P}(v \in U | v \in R) = \frac{1}{2}$. Furthermore, assume equal (relative) group sizes, i.e., $\frac{|Q|}{N} = \frac{|R|}{N} = \frac{1}{2}$. Then, $\frac{T_{R \to Q \cap U} + \beta T_{R \to Q \cap K}}{|Q|} = \frac{\sum_{q \in Q \cap U} \sum_{r \in R} T_{qr}}{N/2} = \frac{\sum_{q \in Q \cap U} \sum_{r \in R} D_{qq}^{-1} A_{qr}}{N/2}$. $D_{qq} = \sum_{u \in Q} A_{qu} + \sum_{v \in R} A_{qv} = \mathbb{P}((u,v) \in E | u \in Q, v \in Q)|Q| + \mathbb{P}((u,v) \in E | u \in R, v \in Q)|R| = \frac{N}{2}[\mathbb{P}((u,v) \in E | u \in Q, v \in Q) + \mathbb{P}((u,v) \in E | u \in R, v \in Q)]$. Therefore, $\frac{T_{R \to Q \cap U} + \beta T_{R \to Q \cap K}}{|Q|} = \frac{\mathbb{P}((u,v) \in E | u \in R, v \in Q)(|Q \cap U| \times |R|)}{[\mathbb{P}((u,v) \in E | u \in Q, v \in Q) + \mathbb{P}((u,v) \in E | u \in R, v \in Q)] \cdot N^2/4} = \frac{\frac{1}{2}|Q| \times |R|}{\left[1 + \frac{\mathbb{P}((u,v) \in E | u \in R, v \in Q)}{\mathbb{P}((u,v) \in E | u \in Q, v \in Q)}\right] \cdot N^2/4} = \frac{1}{2} \cdot \frac{1}{1 + \frac{\mathbb{P}((u,v) \in E | u \in R, v \in Q)}{\mathbb{P}((u,v) \in E | u \in Q, v \in Q)}}$. Similarly, $\frac{T_{Q \to R \cap U} + \beta T_{Q \to R \cap K}}{|R|} = \frac{1}{2} \cdot \frac{1}{1 + \frac{\mathbb{P}((u,v) \in E | u \in Q, v \in R)}{\mathbb{P}((u,v) \in E | u \in R, v \in R)}}$. Thus, $\alpha = \left| 1 - \frac{1}{2} \cdot \frac{1}{1 + \frac{\mathbb{P}((u,v) \in E | u \in R, v \in Q)}{\mathbb{P}((u,v) \in E | u \in Q, v \in Q)}} - \frac{1}{2} \cdot \frac{1}{1 + \frac{\mathbb{P}((u,v) \in E | u \in Q, v \in R)}{\mathbb{P}((u,v) \in E | u \in R, v \in R)}} \right|$. Because $G$ is undirected, $\mathbb{P}((u,v) \in E | u \in R, v \in Q) = \mathbb{P}((u,v) \in E | u \in Q, v \in R)$ and $\mathbb{P}((u,v) \in E | u \in Q, v \in Q) = \mathbb{P}((u,v) \in E | u \in R, v \in R)$.

Ultimately, our theoretical (Section 5) and empirical (Section B.5) characterizations of $\alpha$ and the discrimination risk can be leveraged to audit real-world graph data for structural factors that contribute to the unfairness of mean aggregation feature imputation and ML models applied to the imputed data.

# 6 Fairer graph feature imputation

We propose a simple and effective solution to ensure mean aggregation feature imputation provably has a low discrimination risk, while minimally sacrificing reconstruction error (with respect to the imputation objective). At a high level, we want to constrain the discrimination risk at every iteration $t$ of imputation to be at most $\epsilon$ (i.e., $\forall t \in [0, \infty), \mathcal{R}^{(t)} \leq \epsilon$). We do this by viewing mean aggregation imputation through a gradient descent lens and projecting imputed feature values onto the feasible space of feature values with discrimination risk at most $\epsilon$ at each iteration [82]. We focus on a single feature $i$, but our algorithms can be extended to more features by applying them to each feature separately.

We begin with the case where known feature values remain fixed (i.e., $\beta = 0$). Recall the iterative algorithm for mean aggregation feature imputation when $\beta = 0$:

$$MX^{(t+1)} := \phi(MX^{(t)}) = \begin{bmatrix} 0 & 0 \\ 0 & I_{|U|} \end{bmatrix} TMX^{(t)} + \begin{bmatrix} I_{|K|} & 0 \\ 0 & 0 \end{bmatrix} MX^{(0)}.$$

We see that $\forall t \in [0, \infty), X_K^{(t)} = X_K$. Furthermore, define $\Delta := I_N - M^{-1}TM$. In the case of Feature Propagation, $\Delta = I_N - D^{\frac{1}{2}}(D^{-1}A)D^{-\frac{1}{2}} = I_N - D^{-\frac{1}{2}}AD^{-\frac{1}{2}}$ is the symmetric normalized Laplacian of $G$ [16]. We see that $X_U^{(t+1)} = (I_N - \Delta)_{UU}X_U^{(t)} - \Delta_{UK}X_K$. As discussed in [16], we can view the update $X_U^{(t+1)} := X_U^{(t)} - \gamma(\Delta_{UU}X_U^{(t)} + \Delta_{UK}X_K)$ as an iteration of gradient descent (with step size $\gamma = 1$) on the objective function $\ell(x) = \frac{1}{2}x^T \Delta_{UU}x + X_K^T \Delta_{KU}x + \frac{1}{2}X_K^T \Delta_{KK}X_K$, where $X_K$ is constant. For Feature Propagation, $\ell$ is the Dirichlet energy of $G$ [16]. We now present a theorem that shows how to perform mean aggregation feature imputation with a discrimination risk of at most $\epsilon$ when the known feature values remain fixed.

**Theorem 3 ($\epsilon$-Fair Imputation, $\beta = 0$)** Vanilla mean aggregation feature imputation updates $X_U^{(t+1)} := X_U^{(t)} - \gamma(\Delta_{UU}X_U^{(t)} + \Delta_{UK}X_K) = Z_U^{(t)}$, where $\gamma = 1$. Let $\epsilon$-fair mean aggregation feature imputation instead update $X_U^{(t+1)} := P_W Z_U^{(t)} + P_B$, where:

$$P_W = \begin{cases} I_{|U|}, & \mathcal{R}_K - \epsilon \leq c^T Z_U^{(t)} \leq \mathcal{R}_K + \epsilon \\ I_{|U|} - \frac{cc^T}{c^Tc}, & \text{otherwise} \end{cases}, P_B = \frac{cc^T}{c^Tc} \begin{cases} \mathcal{R}_K - \epsilon, & c^T Z_U^{(t)} < \mathcal{R}_K - \epsilon \\ \mathcal{R}_K + \epsilon, & c^T Z_U^{(t)} > \mathcal{R}_K + \epsilon \\ 0, & \text{otherwise} \end{cases}$$

$$\mathcal{R}_K = \frac{1}{|R|} \sum_{r \in R \cap K} X_r - \frac{1}{|Q|} \sum_{q \in Q \cap K} X_q$$

$$c \in \mathbb{R}^{|U|}, c^T Z_U^{(t)} = \frac{1}{|Q|} \sum_{q \in Q \cap U} Z_q^{(t)} - \frac{1}{|R|} \sum_{r \in R \cap U} Z_r^{(t)}$$

Then, assuming $0 \leq \lambda_{min}(\Delta_{UU}) \leq \lambda_{max}(\Delta_{UU}) < 1$ (where $\lambda_{min}$ and $\lambda_{max}$ are the minimum and maximum eigenvalues, respectively): 1) a unique optimal (with respect to $\ell$) feasible solution $X_U^*$ exists; 2) for fixed step size $\gamma = \frac{1}{\lambda_{max}(\Delta_{UU})}$, $\epsilon$-fair imputation converges as $\|X_U^{(t)} - X_U^*\|_2^2 \leq \left(1 - \frac{\lambda_{min}(\Delta_{UU})}{\lambda_{max}(\Delta_{UU})}\right)^t \|X_U^{(0)} - X_U^*\|_2^2$; 3) for fixed step size $\gamma \leq \frac{1}{\lambda_{max}(\Delta_{UU})}$, $\epsilon$-fair imputation converges to $X_U^*$.

Please refer to Section A.6 for a proof of Theorem 3. The convergence of our solution to the unique optimal (with respect to $\ell$) feasible solution implies that our solution provably has a discrimination risk of at most $\epsilon$ while minimally sacrificing reconstruction error (with respect to the objective). Furthermore, because our solution simply interleaves projections into the mean aggregation imputation framework, it preserves the framework's speed and scalability [16]. We similarly have a solution when $\beta > 0$ (i.e., when the known node feature values do not remain fixed) which we present in Section A.7. Choosing $\epsilon$, due to its uninterpretable nature, can be difficult in practice; we encourage work on making $\epsilon$ more intelligible.

## 7 Experimental results and discussion

We empirically evaluate the fairness and reconstruction error of our solution on various mean aggregation feature imputation algorithms and synthetic and real-world datasets. We find that while our solution yields improved fairness without a significant loss in reconstruction error on the synthetic datasets, there is not an improvement in fairness on the real-world datasets. All our code may be found in the supplementary material.

**Datasets** We construct undirected two-block synthetic networks (SBM) using `StochasticBlockModelDataset` from PyTorch Geometric [83] (where one block corresponds to the marginalized group $Q$ and the other block to the dominant group $R$) with various (relative) group sizes and inter- and intra-link rates (more information in Section B.1). SBM does not have a corresponding task, i.e., the nodes do not have labels. We also use the real-world `Credit defaulter` and `German credit` networks from [29] (there exist limited "natively" graph real-world datasets with sensitive attributes available). The `Credit defaulter` network consists of 30,000 nodes representing individuals, with edges between them indicating similar spending and payment patterns. The corresponding task is to predict whether an individual will default on their credit card payment or not, and the groups are those 25 years old or younger and those above the age of 25 (more information in Section B.1). The `German credit` network comprises 1,000 nodes representing clients in a German bank who are connected if they have similar credit accounts. The corresponding task is to predict whether a client has good or bad credit risk, and the groups are men and women (more information in Section B.1)[3]. We refrain from using the `Recidivism graph` from [29] so as not to support the development of carceral technology [85].

**Protocols and performance evaluation** By default, none of the datasets contain unknown or missing features. Despite the real-world prevalence of missing features, most publicly-available graph datasets inherently do not have missing node features because current graph ML techniques predominantly rely on fully-observable features. Hence, similarly to [16], to simulate diverse scenarios with unknown features, for each group, we independently at random mark node feature values as unknown with a different probability. Nodes, even within the same group, may have different unknown features. In this way, we simulate missing completely at random (MCAR) and missing at random (MAR) on our data. We experiment with all 25 combinations of unknown feature rates of $\{0.1, 0.3, 0.5, 0.7, 0.9\}$ for the groups. We choose this scheme to study the effect of disparate unknown feature rates across the groups, which reflects the real world [17, 18, 19, 20]. Empirically characterizing the real-world distributions of node feature missingness requires further study. We also encourage empirical work on other unknown feature schemes, including missingness based on node degree or marking all or none of the features for each node as unknown.

---

[3]In reality, gender is neither binary nor static, and treating it as such can produce harms [84]

To impute unknown features, we use the vanilla mean aggregation imputation algorithms overviewed in Section 4: Global Mean (GM), Neighbor Mean (NM), Feature Propagation (FP), and Graph Regularization (GR) (with $\beta = 0.25$), and their $\epsilon$-fair counterparts (for $\epsilon \in \{0.0, 0.025, 0.05\}$). For models, we use a linear classifier ($linear$), two-layer MLP ($mlp$), and two-layer Graph Convolutional Network ($gcn$) [2] (more information in Section B.3). We train all models on the imputed data, but validate and test on fully-observed data. Because there are no previous works (to the best of our knowledge) that directly address the unfairness of graph feature imputation, we do not have baselines against which to compare.

To evaluate imputed features for SBM, since we don't have labels, we employ relative reconstruction error **RE** [16]. For the real-world datasets, we consider the test accuracy (**Acc**) of models applied to the imputed data. To evaluate group fairness, we compute the discrimination risk (**DR**) of the imputed data. For SBM, we also train models on the imputed data to predict group membership and calculate the test accuracy of the models on identifying group membership (which we refer to as **MI**) [86, 21]. To evaluate group fairness for the real-world datasets, we use the test statistical parity (**SP**) of the models, defined as $|\mathbb{P}(Z = 1|S = Q) - \mathbb{P}(Z = 1|S = R)|$ [78], and test equalized odds (**EO**), defined as $|\mathbb{P}(Z = 1|S = Q, Y = 1) - \mathbb{P}(Z = 1|S = R, Y = 1)|$ [87]. Please refer to Section B.4 for more details on the metrics. For all metrics, we report the mean and standard error over 5 runs using different random seeds. On each run, a new dataset (in the case of SBM) is generated, new splits are created, and a new model is trained.

**Q1. Does the contraction coefficient $\alpha$ align with the discrimination risk of mean aggregation imputation across graphs with different properties?** Figures 2 to 12 in the Appendix show that for SBM, discrimination risk and $\alpha$ generally have a strong positive association over unknown feature rates, group sizes, and inter-link rate to intra-link rate ratios, which substantiates Theorem 2. This association is weaker for Global Mean and Neighbor Mean. Refer to Section B.5 for more details.

**Q2. Does $\epsilon$-fair mean aggregation imputation (compared to regular mean aggregation imputation) improve the group fairness of a model applied to the imputed data?** Table 1 shows that, for SBM, $\epsilon$-fair mean aggregation imputation achieves comparable reconstruction error to its vanilla counterpart while greatly reducing the discrimination risk and test group membership identification accuracy for all models. As expected, we see that the discrimination risk of $\epsilon$-fair imputation is at most $\epsilon$, and discrimination risk and test group membership identification accuracy are positively associated, which substantiates Theorem 1. Furthermore, the reconstruction error for $\epsilon$-fair FP and GR (which leverage graph structure and are thus more susceptible to graph structural bias) are much lower than that of the naïve $\epsilon$-fair GM (which does not consider graph structure), but fair FP and GR reduce the discrimination risk and test group membership identification accuracy for all models to similar levels as fair GM. However, the test group membership identification accuracy generally decreases less as $\epsilon$ decreases for $mlp$ and $gcn$ than it does for $linear$, which suggests that minimizing the discrimination risk of imputed features is more effective at removing linearly-encoded group membership information than non-linearly encoded information. Furthermore, $\epsilon$-fair feature imputation does not guard against group membership information that $gcn$ learns via graph structure during training. We have similar findings when averaging over different relative group sizes (refer to Table 3) and combinations of inter- and intra-link rates (refer to Table 4).

In contrast, Tables 2 and 6 (in the Appendix) show that, for the real-world datasets, regular mean aggregation imputation and its $\epsilon$-fair counterpart yield comparable test accuracy and statistical parity fairness for all models. In the case of `Credit defaulter`, our solution even appears to exacerbate the unfairness of $gcn$. We find similar results for equalized odds, as shown in Tables 5 and 7. Notably, we were unable to reproduce similar unfairness results to those in [29], even when all features are known. More deeply understanding why our method does not work on the real-world datasets is an important and interesting future work; we would like to analyze the modularity of the real-world networks, as well as the distribution of node degrees, labels, and features across groups to diagnose sources of failure.

## 8    Conclusion

We prove that a higher discrimination risk can amplify the unfairness of a ML model applied to imputed data. We formalize a general graph feature imputation framework called mean aggregation imputation and theoretically and empirically characterize graphs in which applying the framework can yield a high discrimination risk. We propose a simple and effective solution to ensure mean

Table 1: Reconstruction error (**RE**), discrimination risk (**DR**), and test group membership identification accuracy (**MI**) of all models averaged over all 25 combinations of unknown feature rates of $\{0.1, 0.3, 0.5, 0.7, 0.9\}$ for each group in SBM. We use $0.5$ relative group sizes and $0.5$ inter- and intra-link rates.

| Method | $\mathbf{RE}\downarrow$ | $\mathbf{DR}\downarrow$ | $\mathbf{MI}_{linear}\downarrow$ | $\mathbf{MI}_{mlp}\downarrow$ | $\mathbf{MI}_{gcn}\downarrow$ |
|---|---|---|---|---|---|
| 0-Fair GM | $1.21 \pm 0.021$ | $\mathbf{0 \pm 0}$ | $\mathbf{0.602 \pm 0.098}$ | $\mathbf{0.669 \pm 0.019}$ | $\mathbf{0.504 \pm 0.038}$ |
| 0.025-Fair GM | $1.204 \pm 0.021$ | $0.021 \pm 0.002$ | $0.72 \pm 0.087$ | $0.683 \pm 0.01$ | $0.551 \pm 0.069$ |
| 0.05-Fair GM | $1.196 \pm 0.02$ | $0.034 \pm 0.005$ | $0.736 \pm 0.037$ | $0.69 \pm 0.014$ | $0.535 \pm 0.058$ |
| Regular GM | $\mathbf{1 \pm 0}$ | $0.051 \pm 0.015$ | $0.817 \pm 0.02$ | $0.817 \pm 0.013$ | $0.651 \pm 0.089$ |
| 0-Fair NM | $1.19 \pm 0.02$ | $\mathbf{0 \pm 0}$ | $\mathbf{0.599 \pm 0.094}$ | $\mathbf{0.686 \pm 0.021}$ | $\mathbf{0.507 \pm 0.04}$ |
| 0.025-Fair NM | $1.183 \pm 0.02$ | $0.02 \pm 0.002$ | $0.72 \pm 0.084$ | $0.706 \pm 0.013$ | $0.552 \pm 0.059$ |
| 0.05-Fair NM | $1.175 \pm 0.019$ | $0.033 \pm 0.004$ | $0.734 \pm 0.037$ | $0.706 \pm 0.02$ | $0.539 \pm 0.052$ |
| Regular NM | $\mathbf{0.977 \pm 0.002}$ | $0.048 \pm 0.014$ | $0.828 \pm 0.021$ | $0.818 \pm 0.013$ | $0.631 \pm 0.094$ |
| 0-Fair FP | $1.184 \pm 0.020$ | $\mathbf{0 \pm 0}$ | $\mathbf{0.6 \pm 0.096}$ | $\mathbf{0.702 \pm 0.02}$ | $\mathbf{0.505 \pm 0.034}$ |
| 0.025-Fair FP | $1.176 \pm 0.020$ | $0.018 \pm 0.003$ | $0.716 \pm 0.076$ | $0.724 \pm 0.014$ | $0.562 \pm 0.058$ |
| 0.05-Fair FP | $1.169 \pm 0.019$ | $0.025 \pm 0.006$ | $0.72 \pm 0.03$ | $0.723 \pm 0.018$ | $0.531 \pm 0.058$ |
| Regular FP | $\mathbf{0.977 \pm 0.003}$ | $0.028 \pm 0.009$ | $0.814 \pm 0.023$ | $0.817 \pm 0.012$ | $0.612 \pm 0.079$ |
| 0-Fair GR | $1.006 \pm 0.004$ | $\mathbf{0 \pm 0}$ | $\mathbf{0.588 \pm 0.093}$ | $\mathbf{0.713 \pm 0.022}$ | $\mathbf{0.511 \pm 0.039}$ |
| 0.025-Fair GR | $1.005 \pm 0.004$ | $0.023 \pm 0.002$ | $0.757 \pm 0.055$ | $0.741 \pm 0.012$ | $0.577 \pm 0.078$ |
| 0.05-Fair GR | $1.003 \pm 0.004$ | $0.039 \pm 0.006$ | $0.772 \pm 0.027$ | $0.744 \pm 0.016$ | $0.538 \pm 0.066$ |
| Regular GR | $\mathbf{0.977 \pm 0.003}$ | $0.021 \pm 0.007$ | $0.814 \pm 0.024$ | $0.821 \pm 0.01$ | $0.604 \pm 0.08$ |

Table 2: Test accuracy (**Acc**) and statistical parity (**SP**) of all models averaged over all 25 combinations of unknown feature rates of $\{0.1, 0.3, 0.5, 0.7, 0.9\}$ for each group in `German credit`.

| Method | $\mathbf{Acc}_{linear}\uparrow$ | $\mathbf{Acc}_{mlp}\uparrow$ | $\mathbf{Acc}_{gcn}\uparrow$ | $\mathbf{SP}_{linear}\downarrow$ | $\mathbf{SP}_{mlp}\downarrow$ | $\mathbf{SP}_{gcn}\downarrow$ |
|---|---|---|---|---|---|---|
| 0.0-Fair GM | $0.700 \pm 0.006$ | $0.707 \pm 0.003$ | $\mathbf{0.699 \pm 0.002}$ | $0.051 \pm 0.016$ | $0.028 \pm 0.004$ | $0.011 \pm 0.01$ |
| 0.025-Fair GM | $\mathbf{0.705 \pm 0.003}$ | $0.707 \pm 0.003$ | $0.698 \pm 0.002$ | $0.044 \pm 0.012$ | $0.028 \pm 0.007$ | $0.02 \pm 0.026$ |
| 0.05-Fair GM | $0.704 \pm 0.007$ | $\mathbf{0.708 \pm 0.004}$ | $0.697 \pm 0.002$ | $\mathbf{0.034 \pm 0.009}$ | $0.029 \pm 0.003$ | $0.013 \pm 0.005$ |
| Regular GM | $0.701 \pm 0.002$ | $0.708 \pm 0.004$ | $0.699 \pm 0.001$ | $0.043 \pm 0.01$ | $\mathbf{0.025 \pm 0.005}$ | $\mathbf{0.006 \pm 0.005}$ |
| 0.0-Fair NM | $0.699 \pm 0.005$ | $0.706 \pm 0.003$ | $0.697 \pm 0.003$ | $0.053 \pm 0.015$ | $\mathbf{0.033 \pm 0.007}$ | $\mathbf{0.007 \pm 0.006}$ |
| 0.025-Fair NM | $\mathbf{0.7 \pm 0.006}$ | $0.706 \pm 0.003$ | $0.697 \pm 0.002$ | $\mathbf{0.041 \pm 0.009}$ | $0.037 \pm 0.007$ | $0.015 \pm 0.013$ |
| 0.05-Fair NM | $0.7 \pm 0.006$ | $0.706 \pm 0.003$ | $0.697 \pm 0.002$ | $0.046 \pm 0.013$ | $0.033 \pm 0.005$ | $0.01 \pm 0.003$ |
| Regular NM | $0.7 \pm 0.003$ | $\mathbf{0.708 \pm 0.001}$ | $\mathbf{0.698 \pm 0.001}$ | $0.044 \pm 0.02$ | $0.034 \pm 0.007$ | $0.016 \pm 0.007$ |
| 0.0-Fair FP | $0.694 \pm 0.021$ | $\mathbf{0.713 \pm 0.011}$ | $\mathbf{0.704 \pm 0.007}$ | $\mathbf{0.012 \pm 0.012}$ | $0.025 \pm 0.025$ | $0.019 \pm 0.039$ |
| 0.025-Fair FP | $\mathbf{0.708 \pm 0.012}$ | $0.706 \pm 0.012$ | $0.689 \pm 0.027$ | $0.024 \pm 0.028$ | $0.026 \pm 0.026$ | $0.026 \pm 0.057$ |
| 0.05-Fair FP | $0.702 \pm 0.03$ | $0.706 \pm 0.011$ | $0.708 \pm 0.046$ | $0.078 \pm 0.01$ | $\mathbf{0.023 \pm 0.026}$ | $\mathbf{0 \pm 0}$ |
| Regular FP | $0.7 \pm 0.024$ | $0.708 \pm 0.012$ | $0.698 \pm 0.01$ | $0.063 \pm 0.069$ | $0.03 \pm 0.02$ | $0.007 \pm 0.01$ |
| 0.0-Fair GR | $0.698 \pm 0.005$ | $\mathbf{0.703 \pm 0.004}$ | $0.698 \pm 0.001$ | $0.04 \pm 0.02$ | $\mathbf{0.021 \pm 0.003}$ | $0.006 \pm 0.007$ |
| 0.025-Fair GR | $\mathbf{0.702 \pm 0.004}$ | $0.702 \pm 0.002$ | $\mathbf{0.7 \pm 0.001}$ | $0.041 \pm 0.014$ | $0.025 \pm 0.002$ | $0.008 \pm 0.01$ |
| 0.05-Fair GR | $0.699 \pm 0.004$ | $0.703 \pm 0.003$ | $0.699 \pm 0.003$ | $\mathbf{0.034 \pm 0.01}$ | $0.024 \pm 0.004$ | $\mathbf{0.005 \pm 0.005}$ |
| Regular GR | $0.697 \pm 0.003$ | $0.703 \pm 0.003$ | $0.7 \pm 0.001$ | $0.038 \pm 0.017$ | $0.027 \pm 0.005$ | $0.01 \pm 0.009$ |

aggregation-imputed features provably have a low discrimination risk, while minimally sacrificing reconstruction error (with respect to the imputation objective).

Our analysis and solution, like many fair ML algorithms, assume that groups are discrete and that group membership is known and static, which is not true in reality [82, 84, 26]. Furthermore, we don't consider fairness at the intersections of different groups [88, 89], or operationalizations of fairness beyond the parity of two non-overlapping groups [90]. Furthermore, while fairness is often framed as sufficient for the creation of ethical systems, this is often not the case. For instance, $\epsilon$-fair mean aggregation imputation may be used to train a "fairer" model that diversifies news recommendations to social media users [7], but this model could recommend hostile or intolerant news sources to LGBTQIA+ users and cause psychological harm [82].

## Acknowledgments and Disclosure of Funding

This work was partially supported by NSF III-1705169, NSF 1937599, NSF 1927554, NSF 2119643, Okawa Foundation Grant, Amazon Research Awards, Cisco research grant USA000EP280889, Sloan Research Fellow, Picsart Gifts, and Snapchat Gifts.

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
