# OpenReview forum: "On the Discrimination Risk of Mean Aggregation Feature Imputation in Graphs"
_NeurIPS.cc/2022/Conference — NeurIPS 2022 Accept_

### Official Review · Reviewer_dRAB · 2022-07-17

**Rating:** 6
**Confidence:** 3
**Soundness:** 3 good
**Presentation:** 2 fair
**Contribution:** 2 fair

**Summary:**

This paper studies the effect of mean aggregation feature imputation in graphs over marginalized and dominant groups. The main conclusion is that mean aggregation imputation may harm fairness between those two groups. To alleviate this unfairness, the paper proposes a corrected graph feature imputation algorithm that considers the feature unfairness in their updates.

**Questions:**

1. The contraction coefficient alpha is hard to interpret in Theorem 2. As explained in line 209, T is a right-stochastic matrix without an explicit form. So it is hard to judge the scale of alpha.
2. Theorem 2 can be explained as the discrimination risk is upper-bounded by the initial difference between two instances either within the marginalized group or within the dominant group. That seems not to be a good sign since diversity may show up within each group. Does that mean the diversity among data also harms fairness?
3. In Theorem 3 it seems that epsilon-fair computation does not affect the convergence rate of X_U, since the convergence rate relies on eigenvalues of Delta_{UU} only, and the graph topology is not modified. Does that mean epsilon-fair computation can do as well as vanilla mean aggregation feature imputation given enough data?

**Ethics Review Area:**

["I don’t know"]

**Limitations:**

This paper studies mean aggregation feature imputation and its harm to fairness. Several things are not covered. First, the analysis of the classification model (X->Y) seems over-simplified. Second, the paper considers the simplest case that the dataset only consists of two discretized, non-overlapping groups.

**Strengths And Weaknesses:**

Strengths:
1. This paper articulated the source of unfairness can propagate from mean aggregation feature imputation to discrimination in model predictions. The propagation effect can break into two parts. First, the discrimination can propagate from input to output (Theorem 1). Second, the iterated mean aggregation feature imputation algorithm can exaggerate the input discrimination (Theorem 2).
2. The paper proposes an algorithm to compensate for the discrimination across groups in each iteration. The algorithm can provably satisfy epsilon-fairness.
3. Empirical result shows a clear tradeoff between utility and fairness, which is expected.

Weaknesses:
1. It is hard to get the intuition behind the fair graph feature imputation algorithm.

---

> ### Author Response · Authors · 2022-08-02
> **Response to Reviewer dRAB**
>
> Thanks for your feedback and insightful questions! We would like to address your questions below:
> - (Question 1) We provide interpretations for the contraction coefficient in terms of graph properties in Section A.5.
> - (Question 2) Yes, a large initial maximal deviation in feature values within a group can harm fairness, but a large initial deviation does not necessarily entail diversity. For example, suppose a few nodes in a group have a low initial feature value but many more nodes have a much higher initial feature value (i.e., large initial difference). (Depending on definition, this might not be considered “diverse.”) Then, after mean aggregation, the feature values for all the nodes in the group may be higher on average than they were initially, and more distinct on average from the node feature values in the other group. This would contribute to a higher discrimination risk.
> - (Question 3) Yes, epsilon-fair imputation can do just as well (in terms of convergence rate) as vanilla feature imputation.
> - (Limitations) Could you further elaborate on which parts of the analysis seem over-simplified? We agree with the limitation about “non-overlapping groups,” which pervades the fairness literature. However, if there are more than two groups, Theorem 2 can be extended by considering all pairs of groups. Additionally, our fair feature imputation algorithm could be generalized to multiple groups by considering all pairs of groups and projecting onto the intersection of their feasible spaces at each iteration.

---

### Official Review · Reviewer_QttQ · 2022-07-17

**Rating:** 6
**Confidence:** 4
**Soundness:** 3 good
**Presentation:** 4 excellent
**Contribution:** 4 excellent

**Summary:**

The authors consider the problem of fairness when imputing features in a group setting where on average there is higher difference to ground truth among the marginal groups vs the dominant groups. Experiments are shown on both synthetic and real-world fairness-utility tasks.



**Questions:**

(A) If we do not the group information of the applicants/samples, how does the algorithm work? Do you use the group information only for the discrimination risk but not for the feature imputation step?

(B) How do we use the correlated features in the imputation process since occupation/title is correlated with salary?

**Limitations:**

However, most of the results on real-world datasets are not providing any of the promised lift in terms of fairness.

**Strengths And Weaknesses:**

The paper is well-written with both theoretical results in main paper as well as extensive experiments. In particular, the limitations of the approach on real-world datasets is clearly highlighted.

The setting is interesting because fairness on graphs has not been theoretically studied given the practical importance of the problem. The authors propose a discrimination measure which is broadly applicable, not limited by models or graphs, and show that it closer to the statistical parity often used in fairness literature.

I checked over the main Theorems and proof sketches but did not look over the supplementary in detail.

---

> ### Author Response · Authors · 2022-08-02
> **Response to Reviewer QttQ**
>
> Thanks for your helpful comments! We would like to address your questions below:
> - (Question A) While vanilla feature imputation does not require knowing group membership, both computing the discrimination risk and our proposed fair feature imputation algorithm require being able to fully observe group membership.
> - (Question B) Incorporating correlations between features in the imputation process is a promising direction of research. Unfortunately, this is out of the scope of this paper, as we are concerned with improving the fairness of previously proposed imputation algorithms (e.g., [1]), which don’t consider feature correlations.
>
> [1] https://openreview.net/forum?id=tx4qfdJSFvG

---

### Official Review · Reviewer_zs22 · 2022-07-18

**Rating:** 4
**Confidence:** 3
**Soundness:** 2 fair
**Presentation:** 2 fair
**Contribution:** 3 good

**Summary:**

This paper discusses some of the problems related to fairness that can occur when a machine learning model is applied to graph data with missing values. Fairness is considered concerning two groups, a marginalized group ($Q$) and a dominant group ($R$). It seeks to derive theoretical results relating the Total Variation Distance between the conditional prediction distribution given imputed features and ground truth features with a measure the authors term the Discrimination Risk---defined to be the minimum  L1 distance between the mean (imputed) feature for marginalized and non-marginalized groups (with the minimum taken across feature dimensions). The paper then investigates feature imputation in graph data, aiming to derive an algorithm for iteratively imputing missing data while controlling the Discrimination Risk. Performance evaluation is then done on synthetic Stochastic Block Model (SBM) data and on credit datasets.


**Questions:**

-I think, in line 172 of p. 4, "$a$" should read "$a_i$".

-In Lemma 1, it would appear difficult to interpret the Total Variation distance inequalities, as lefthand and righthand side quantities are defined over a different support ($Z$ (low dimensional) and $X$ (presumably high dimensional).


**Limitations:**

- One limitation is that, for a given graph imputation model, it is unclear what a reasonable baseline DR would be (a problem made more pronounced by the fact that it depends on the scale of the covariate data). Therefore, the choice of $\epsilon$ in the algorithmic component is hard to make in practice and a function of human judgement.

- Knowledge of marginal and dominant groups is required a priori. While in some situations, there may be little debate about this fact, in other situations, the question of who is in fact marginalized can be politically contentious or simply difficult to make.

**Strengths And Weaknesses:**

The overall idea for this paper is strong. Missing data is certainly a prevalent problem and to my knowledge there is not much work connecting the imputation and fairness in the graph context. Therefore, solutions to the problem identified by the paper would in my view be valuable. The evaluation procotol is reasonably strong, and the theoretical results intriguing.

Nevertheless, there are several things that I currently perceive to be weaknesses with the paper.

The notion of Discrimination Risk (DR) is central to the paper. As written, the theoretical results related to DR depend on covariates being bounded/continuous. Moreover, it is unclear whether the DR value is used for mathematical convenience or because it captures a true quantity of interest related to fairness. For example, the DR value is defined in Equation 1 to be the minimum L1 distance across dimensions between the mean (imputed) feature for marginalized and non-marginalized. Adding additional random covariates would bring this measure to 0 (i.e., the expected value of the minimum of a set of random variables scales with n or in this case d). Also, if there is one dimension where marginalized and dominant groups are equal in the covariate space, the measure is also 0. It could be the authors meant to write max instead of min, as in Equation 3, an apparently different definition of DR is introduced. (One other point is that the DR measure assumes the existence of exclusive marginalized and dominant groups viewed asymetrically, conversely, the absolute value in Equation 1 will penalize deviations symmetrically.)

The confusion related to DR is also related to a more general point about the paper presentation. Parts of the paper are hard to follow. Some of this would be improved with a more consistent emphasis in the paper. The paper starts off with a fairly clear introduction regarding graph imputation, but third section makes no real mention of graph data; graphs are then re-introduced in the fourth section. Some of the claims in the abstract are also somewhat too strong or potentially misleading in my view (especially the "minimally sacrificing utility" point; this isn't false, but the "utility" meant here is not related to social utility as sometimes understood, but to the specific objective function in line 265 of p. 7).

Regarding the simulation design -- First, the two real-world datasets are not "natively" actually network or graph data. Edges between units simply represent unit-level similarity (i.e. do units share spending patterns?; do units have similar credit accounts?). This limits our ability to access performance in actual network data where relations between units are not just deterministic functions of the pairwise unit-level covariates.

Second, the critical contribution that the papers seeks to make is to devise strategies for missing data imputation in graph data. Therefore, it would seem appropriate to have the missingness mechanism be correlated with graph specific quantities (e.g., non-central nodes are more likely to have missingness or something of the sort).

---

> ### Author Response · Authors · 2022-08-02
> **Response to Reviewer zs22**
>
> We greatly value your detailed and helpful feedback! We would like to address some of the weaknesses and questions you raised:
> - Regarding random covariates, you are correct, we meant to write max instead of min. This has been fixed in the revised draft.
> - The discrimination risk captures a true quantity of interest related to fairness. For example, as we note in the paper on lines 181-182, our definition of discrimination risk may be alternatively viewed through the lens of the Average Treatment Effect studied in counterfactual fairness and causality (where group membership is the treatment and features are the outcome). Furthermore, similar quantities have been proposed in [1] and [2].
> - Regarding the paper presentation, we appreciate your helpful comments. We had hoped to first present the discrimination risk as a general, domain-agnostic quantity before exploring it in the context of graphs, but we understand how this is confusing. We will more consistently emphasize central topics in our final draft.
> - Your comment about the polysemy of “utility” is important. We have appropriately replaced “utility” with “reconstruction error” and “accuracy” in the revised draft.
> - Concerning experiments, most publicly-available graph datasets used for evaluation inherently do not have missing features because current graph machine learning techniques predominantly rely on fully-observable features. This is why we simulate missing node features like [3]. (However, In our final draft, we will include results for experiments wherein feature missingness depends on structural properties, e.g., node degree, node centrality.) Furthermore, there are only a few benchmark graph datasets with sensitive attributes available [4]. Hence, it was difficult for us to identify graph datasets that had both (1) naturally missing node features, and (2) sensitive attributes available.
> - (Question 1) This is no longer an issue in the revised draft.
> - (Question 2) The support should not influence one’s interpretation of the inequality. $d_{TV} (\cdot, \cdot)$ only requires that its two arguments have the same support. Because $d_{TV}$ outputs the largest possible difference between the probabilities that the two distributions can assign to the same event, the inequality can be viewed as a comparison of the differences in assigned probabilities.
> - (Limitations) We agree with the limitations, which pervade the fairness literature.
>
> [1] https://openreview.net/forum?id=xgGS6PmzNq6
>
> [2] https://arxiv.org/abs/2109.10431
>
> [3] https://openreview.net/forum?id=tx4qfdJSFvG
>
> [4] https://arxiv.org/abs/2102.13186

---

### Official Review · Reviewer_Ne2Y · 2022-07-20

**Rating:** 4
**Confidence:** 4
**Soundness:** 2 fair
**Presentation:** 3 good
**Contribution:** 3 good

**Summary:**

The authors studied the effect of graph feature imputation on the fairness of models. Specifically, they defined the discrimination risk (measures how much features differ between groups) and showed that statistical parity is upper-bounded by the discrimination risk. Based on a general graph feature imputation framework that includes several existing imputation methods as examples, the authors analyzed the cases in which applying this imputation framework leads to higher discrimination risk and proposed a $\epsilon$-fair imputation method for low discrimination risk. The proposed method empirically showed improvement on synthetic datasets but not on real-world credit datasets.

**Questions:**

Please see the weaknesses above.

**Limitations:**

Yes, the authors discussed the limitation.


**Strengths And Weaknesses:**

Strengths:
* The authors analyzed the effect of graph feature imputation on the fairness of models, which is an important problem and often encountered in practice.
* The authors theoretically (Theorem 2) and empirically (Appendix) investigated how the graph properties affect $\alpha$, which helps interpret the impact of existing imputation methods on the statistical parity.
* The authors also proposed a solution to control discrimination risk under the general mean aggregation feature imputation framework.

Weaknesses:
* Could the authors explain in which cases statistical parity is preferable to predictive parity and why they focused on statistical parity in this problem?
* It seems to me that all the analysis is limited to two groups starting from the definition of discrimination risk. How generalizable the current results are to multiple groups?
* It is not clear to me the impact of graph properties on the proposed $\epsilon$-fair imputation method in Theorem 3.
* Empirical evidence of the effectiveness of the proposed solution is limited.
* It would be better to shorten some long sentences for easier interpretation, such as lines 5-9.

---

> ### Author Response · Authors · 2022-08-02
> **Response to Reviewer Ne2Y**
>
> Thanks for your valuable feedback! We would like to address your questions below:
> - (Question 1) We erroneously used the term “predictive parity” instead of “equal opportunity;” we have corrected this in the revised draft. If we consider the automated candidate screening example from the paper, statistical parity (3.1.1 in [1]) would imply that all candidates have an equivalent opportunity to pass screening regardless of group membership, while equal opportunity (3.2.3 in [1]) would mean that, regardless of group membership, candidates are classified with an equivalent accuracy by the automated screening model. However, we would likely prefer statistical parity to equal opportunity when ground-truth labels are highly associated with group membership. For example, if the candidate screening system uses hiring data, because of systemic sexism, being a woman would be associated with a lower chance of being hired.
> - (Question 2) Yes, the analysis in the paper only considers two groups. However, if there are more than two groups, Theorem 2 can be extended by considering all pairs of groups. Additionally, our fair feature imputation algorithm could be generalized to multiple groups by considering all pairs of groups and projecting onto the intersection of their feasible spaces at each iteration.
> - (Question 3) The graph properties (discussed in Section A.5) are not directly related to Theorem 3. Theorem 3 seeks to minimize the empirical discrimination risk, while Theorem 2 explores which graph properties could increase the discrimination risk.
> - (Question 5) We have broken up Lines 5-9 for easier interpretation.
>
> [1] https://fairware.cs.umass.edu/papers/Verma.pdf

---

### Official Review · Reviewer_qNJZ · 2022-07-21

**Rating:** 4
**Confidence:** 4
**Soundness:** 3 good
**Presentation:** 3 good
**Contribution:** 3 good

**Summary:**

This paper provides theoretical analysis on the estimation error of fairness, in the setting where missing data is present. In particular, the authors consider the “discrimination risk” as the fairness metric. The paper also includes some numerical experiments to validate the theoretical findings.

**Questions:**

Please refer the Weaknesses part.

**Strengths And Weaknesses:**

Weaknesses: 1)The authors claim that the solution they provided have a low discrimination risk (while minimally sacrificing utility) and improve the fairness of models. In addition to the fairness, the underline data distribution should be hold after imputation, which directly affect the effectiveness of the models. 2)While the theoretical results can be useful in understanding the behavior of fairness estimation under different settings, I am not sure if they will have much practical impact. 3)In Theorem 3, R_K is a scalar while Z_U is a vector or matrix, that means R_K and Z_U cannot be compared, please explain the corresponding constraint conditions in P_W and P_B. 4)There are three missing data contexts: missing completely at random (MCAR), missing at random (MAR) and missing not at random (MNAR), does the findings in this paper be adequate for all the three missing data contexts?

Overall, the paper is well-written, and the contributions and limitations are clearly described. While the theoretical perspective for fairness estimation with missing data for graph data is novel, I am leaning towards reject due to concerns about practical impact, as mentioned above.

---

> ### Author Response · Authors · 2022-08-02
> **Response to Reviewer qNJZ**
>
> We greatly appreciate your helpful review! We would like to address the weaknesses you highlighted:
> - (Weakness 1) With regards to “the underline data distribution should be hold after imputation,” this is not necessarily the case. Parity-based fairness interventions often reduce the distance between the underlying data distributions for different groups, thereby inherently modifying the distributions.
> - (Weakness 2) Concerning “I am not sure if they will have much practical impact,” our theoretical and empirical characterization of the contraction coefficient $\alpha$ (Section A.5, Section B.5) can be leveraged to audit real-world graph data for structural factors that contribute to bias in models trained on the data.
> - (Weakness 3) In Theorem 3, $Z_U$ is a vector; this is because we consider the setting of one feature per node. $c^T$ is also a vector (defined at the bottom of Theorem 3). Therefore, $c^T Z_U$ is a scalar and can be directly compared to ${\cal R}_K$.
> - (Weakness 4) The theoretical results in our paper apply to all three missing data contexts. However, for our empirical results, we simulate missing completely at random (MCAR) and missing at random (MAR) on our data by randomly marking node features as missing with rates that depend on the group to which the node belongs. In future experiments, we can also explore the MNAR setting by, for example, marking node features as missing based on node degree.

---

### Official Review · Reviewer_6GqN · 2022-07-31

**Rating:** 6
**Confidence:** 3
**Soundness:** 3 good
**Presentation:** 3 good
**Contribution:** 2 fair

**Summary:**

[Last minute review invitation so limited time]

This paper highlights the concern of unfair discrimination caused by imputation of missing feature values in graphs. A measure of this discrimination risk is formed, based on the difference of the expected value of the feature values between a minority / marginalized group and a dominant group. This work goes further by forming a generalised method of mean aggregation feature imputation in order to analyse the effects of imputation in theory and in practise with a number of approaches that are special cases of the generalised method, and introduce a modified form to address the discrimiation risk within an epsilon bound. They show that the theory supports their claims. Empirical evaluation on simulated stochastic block model (SBM) data also supports their cliams and their method to address the problem behaves as expected. However, on real data the results are not as expected, leading to future research to investigate.

**Questions:**

Does [15] do node feature imputation, I don't recall that paper discussing this?

It's not completely clear how the Mean Aggregation Feature method can encompass the Graph Regularisation, is it that T can be replaced with the weighted adjcency matrix after some normalisaiton to be a stochastic matrix, maybe good to clarfiy briefly in the main text?

For Theorem 2, is the maximal deviation the deviation from the mean of the group value or the maximum deviation between any two feature values, I believe it is the later but good to clarify in the main text?

To clarify the conclusion on connections within groups (intra) and between groups (inter), if there are more intra-connections than inter-connections the discrimintation risk is higher?

None of the example datasets have missing features; can you comment on why not as we would hope to see this problem in real world example data. Furthermore, by randomly removing features, how close to reality is this, what kind of distributions do missing features have in the real world?


**Limitations:**

The paper is good at showing the limitations, where by the experiments on real-world data did not show improvements. A hypothesis for this is put forward - that there is discrimination existing in the known labelled data and so it is not able ot improve the discrimination risk. However, this work seems to be heading in a very good direction to address this discrimination risk issue, however, the more interesting question arising is why this did not work on the real data, and can this be more deeply investigated? So why this didn't work on the real data is a very interesting question that is raised in this work.

**Strengths And Weaknesses:**

Originality
========
Strength. In the growing area of graph neural networks this paper highlights a potential risk of discriminating against minority / marginalized groups when imputing feature values for graph nodes.
Weakness. Although in theory this is a genuine problem, the paper does not strongly support that this is a real-world phenomena nor that their method can address this.

Quality and Clarity
===============
Strength. A very well written paper. Easy to follow, lots of relevant citations, well writtne, clear mathematics explained clearly.
Weakenss. Deeper into the paper it was tricky to follow some of the claims and quite often the reader is pointed to the appendix for further details. This is okay as long as it is possible to grasp the intuition of the claim from the main text in my opinion.

Significance
==========
Strength. The problems with discrimination for real world applications of machine learning like credit risk is genuine and of high importance in my opinion, and very well motivated in this paper. I would even add that in my countries this type of discrimination is even illegal, like in the US they have disparate impact labour laws to protect against this type of discrimination.
Weakness. Although it feels as though the idea of feature imputation causing this problem is logical, as explained with illustrations and supported by theory and experiments on simulated data, none of the real-world data they experimented on had missing feature values, which leads us to believe that this is not a common problem in reality, and furthermore the experiments on the real-world data were not clear that there was any improvements when trying to reduce the discrimination risk by removing feature values (to simulate missing feature values) and then using their method to impute them. Although it was commented that the author(s) believe this is likely due to the large amount of discrimination already in the known features, this was not supported by investigation, which maybe could reveal something more about this?

---

> ### Author Response · Authors · 2022-08-02
> **Response to Reviewer 6GqN**
>
> Thank you so much for your helpful feedback! We would like to address your questions:
> - (Question 1) Yes, you are correct, the paper does not explicitly concern node feature imputation. The matrix completion method that the authors propose can be leveraged to impute node features (e.g., if the nodes are users and the features are items). However,  to avoid confusion, we have removed [15] in the revised draft.
> - (Question 2) Your intuition is correct. The proof for how Mean Aggregation encompasses Graph Regularization is nearly identical to the proof for Feature Propagation, with the exception of $\beta \in (0, 1]$.
> - (Question 3) The maximal deviation is the maximum deviation between any two feature values in the same group.
> - (Question 4) Yes, you are correct, more intra-connections than inter-connections yields a higher discrimination risk.
> - (Question 5) Most publicly-available graph datasets inherently do not have missing node features because current graph machine learning techniques predominantly rely on fully-observable features. We believe this is the same reason that [1] similarly simulates missing features. Additionally, there are only a few graph datasets with sensitive attributes available. Hence, it was difficult for us to identify graph datasets that had both (1) naturally missing node features, and (2) sensitive attributes. However, as we motivate in Lines 23-25, features are indeed often missing in real-world data, like social and credit networks, “for privacy reasons, as a consequence of exclusionary data collection practices, or due to the high expenses involved in feature annotation [12, 13, 14].” Empirically characterizing the real-world distributions of node feature missingness would require further study.
> - (Limitations) We agree that more deeply understanding why our method does not work on real-world data is an important and interesting research direction. We would like to analyze the modularity of the real-world networks, as well as the distribution of node degrees, labels, and features across groups to diagnose sources of failure.
>
> [1] https://openreview.net/forum?id=tx4qfdJSFvG

---

### Author Response · Authors · 2022-08-02
**General Response to Reviewers and Fixed Loose Bounds in Theorem 1**

We deeply appreciate the detailed and constructive feedback from all the reviewers! Besides addressing reviewers' comments, we also addressed a loose bound issue in Theorem 1 that we discovered after submission. The previous upper bounds in Theorem 1 were vacuously true because the maximum possible total variation distance between any two distributions is 1. For general distributions, even matching an infinite number of moments is not sufficient to bound their distance [1]. Hence, we have updated Theorem 1 in the revised version of the paper we uploaded, wherein we now leverage stronger generative assumptions to prove both lower and upper bounds for the total variation distance in terms of the discrimination risk.

[1] https://www.jstor.org/stable/2685775

---

### Author Response · Authors · 2022-08-06
**Author-Reviewer Discussion Period**

Dear Reviewers,

We are truly thankful for your valuable feedback! We have tried to address all of your concerns in our responses. As the author-reviewer discussion period will end soon (until Aug. 9), we would love to hear if you still have any concerns and we are more than happy to discuss them.

---

### Meta-Review · Area_Chair_xBdS · 2022-08-28

**Recommendation:** Accept
**Confidence:** Certain

**Metareview:**

The work discusses some of the problems related to fairness that occur when a machine
learning model is applied to data with missing values in graphs. The authors propose a methodology to
compensate for the discrimination across groups.

All the reviewers and the AC agree that the paper overall idea of the paper is strong and very interesting.
The paper is extremely well-written, easy to follow and contirbutions and limitation well described.
The main concern with the paper is its practicality as results on real-world datasets are not providing
any promising lift in terms of fairness. This is also acknowledged as a feature direciton by the authors in the rebuttal.
The AC believes that this is a promsing and impactful line of work and will ignite interesting discussions in the NeurIPS community.
As another reviewer pointed out, there is not much work connecting the imputation and fairness in the graph context.
Acceptance is recommended.

**Award:**

No

---

### Decision · Program_Chairs · 2022-09-14

Accept